# Using Egg Parasitoids to Manage Caterpillars in Soybean and Maize: Benefits, Challenges, and Major Recommendations

**DOI:** 10.3390/insects15110869

**Published:** 2024-11-05

**Authors:** Adeney de F. Bueno, Weidson P. Sutil, M. Fernanda Cingolani, Yelitza C. Colmenarez

**Affiliations:** 1Empresa Brasileira de Pesquisa Agropecuária—Embrapa Soja, Londrina 86085-981, PR, Brazil; 2Programa de Pós-Graduação em Entomologia, Universidade Federal do Paraná (UFPR), Curitiba 91531-980, PR, Brazil; plauter80@gmail.com; 3Centro de Estudios Parasitológicos y de Vectores (CEPAVE), Consejo Nacional de Investigaciones Científicas y Técnicas (CONICET) and Universidad Nacional de La Plata (UNLP), Boulevard 120 s/n, Av. 60 and Calle 64, La Plata 1900, Buenos Aires, Argentina; fernandacingolani@cepave.edu.ar; 4Centre for Agriculture and Bioscience International (CABI) Latin America and Fundação de Estudos e Pesquisas Agrícolas e Florestais (FEPAF)—Avenida Universitária, 3780, Botucatu 18610-034, SP, Brazil; y.colmenarez@cabi.org

**Keywords:** *Telenomus remus*, *Trichogramma pretiosum*, biological control

## Abstract

Egg parasitoids, such as *Trichogramma pretiosum* and *Telenomus remus*, offer an effective tool to manage lepidopterous in soybean and maize. *Trichogramma pretiosum* is already registered for commercial releases in Brazil, while *Te. remus*, which is more efficient against *Spodoptera* spp., is not yet registered. The registration of *Te. remus* would be highly beneficial for controlling pests from the genus *Spodoptera* in Brazil. This review article discusses major recommendations for farmers to efficiently adopt Augmentative Biological Control programs with *Tr. pretiosum* and *Te. remus.* Because their successful use requires proper implementation, these parasitoids should be adopted with different integrated pest management (IPM) practices, including selective pesticide use in addition to other less harmful pest control tools, such as OGMs plants or other biocontrol technology in order to maximize the effectiveness of releasing the egg parasitoids.

## 1. Introduction

Soybean and maize are among the largest and most important crops worldwide. Since the early 1990s, soybean–maize succession has been the most common agricultural production system in Brazil, with soybean being cultivated in the summer, succeeded by maize in the fall/winter [1]. However, this continuous cropping system increases pest outbreaks due to the continuous availability of host plants [2,3], also known as the “green-bridge”. This particularly favors polyphagous species such as *Spodoptera frugiperda* (JE Smith, 1797) (Lepidoptera: Noctuidae) [4], which can feed on 353 different plants belonging to 76 botanical families [5].

Injury by pests can occur throughout the whole plant’s development, reducing the yield of both soybean [6] and maize [7] unless properly managed [8]. In addition to *S. frugiperda*, other Lepidoptera species, especially from the families Noctuidae and Erebidae, are among the key pests requiring constant management to protect crop yield [3,4,5,6,7,8,9]. Traditional chemical insecticides are often the first line of defense adopted by farmers to control pest outbreaks [10]; however, when misused, insecticides negatively impact human health and the environment [11,12]. The overuse of chemical insecticides, especially the most harmful ones, has reduced natural biological control [13], created resistant insect pests in both soybean [14] and maize [15], favored pest resurgence, and led to outbreaks of secondary pests [3], among other negative side-effects.

Alternatively, the development and adoption of genetically modified soybean and maize expressing insecticidal proteins from *Bacillus thuringiensis* (*Bt*) have revolutionized integrated pest management (IPM) programs worldwide [16,17]. Particularly due to its high efficacy against targeted pests associated with its easiness of adoption, the area of *Bt* crops worldwide has grown rapidly since the launch of the technology. This has effectively controlled the target pests [9,18] and reduced the use of insecticides, consequently favoring the conservation of natural antagonists [19,20,21,22]. Despite the significant benefits of adopting *Bt* crops, such technology brings high risks of pest resistance if not properly managed [23]. Insect resistance management in *Bt* crops relies on the cultivation of refuge areas containing non-*Bt* plants [24]. Refuge areas supply susceptible individuals to minimize non-random mating among the rare resistant homozygotes that survive on *Bt* plants, ensuring that the next generation consists of insects susceptible to the high-dose-*Bt* plants [25,26]. Although it may vary depending on the pest species and crop, it is generally recommended to cultivate 20% of non-*Bt* soybean as a structured refuge [27]. A structured refuge in maize varies from 10% (Brazil) to 20% (USA) of non-*Bt* maize [23].

*Bt* crops are widely adopted worldwide, increasing from one million hectares in 1996 to 109 million hectares in 2019 (109% increase) [28]. Considering only soybean and maize, the most recent published data include 84% adoption of *Bt* maize in the United States in 2022 [29,30] and 74% adoption of *Bt* soybean in Brazil in 2019/2020 (Figure 1A) [31]. Despite the recommended 80% limit (taking the Brazilian average into consideration; Figure 1A), at the state level, Bahia (91% of adoption of *Bt* soybean), Maranhão (86% of adoption of *Bt* soybean), Mato Grosso do Sul (88% of adoption of *Bt* soybean), and São Paulo (87% of adoption of *Bt* soybean) already exceeded the 80% limit for *Bt* soybean adoption (Figure 1B).

This low compliance with refuge adoption has triggered the selection of resistance pest populations [31]. Consequently, since the 2019/20 crop season, unexpected defoliation of *Bt* soybean (expressing only Cry1Ac) caused by *Rachiplusia nu* (Guenée, 1852) (Lepidoptera: Noctuidae) [17], as well as damage by *Crocidosema* sp. (Lepidoptera: Tortricidae) [32], has been reported in Brazil [31,32,33]. These were the first confirmed cases of resistance of Lepidoptera species to *Bt* soybean (expressing only Cry1Ac) [17].

After years of a low Lepidoptera population, suppressed by Cry1Ac adoption in Brazil, the occurrence of those first Cry1Ac-resistant species and Cry1Ac-tolerant species (*Spodoptera* spp.) has brought back the spraying of traditional chemical insecticides on soybean [8,9,10,11,12,13,14,15,16,17,18,19,20,21,22,23,24,25,26,27,28,29,30,31,32,33,34]. Unfortunately, such sprays have been carried out even when injuries were lower than the economic thresholds of 30% defoliation in the soybean vegetative stage or 15% defoliation in the soybean reproductive stage for defoliators [35] or 50% injured plants for *Crosidosema* sp. [36]. This increased insecticide use jeopardizes one of the most important benefits of the adoption of *Bt* technology, which is the reduced use of chemical insecticides.

Therefore, a more sustainable alternative to mitigating the increase in Lepidoptera pests that are resistant or tolerant to *Bt* proteins is the adoption of Augmentative Biological Control (ABC) [37]. Egg parasitoids could be one of the most promising alternatives to manage lepidopteran pests [38] and other insect pests, such as stink bugs [39], because they control these pests in their first stage of development (egg) before any damage is caused to plants [40].

Promising parasitoids for managing economically important caterpillars in soybean and maize include *Trichogramma pretiosum* (Riley, 1879) (Hymenoptera: Trichogrammatidae) [41,42] and *Telenomus remus* Nixon, 1937 (Hymenoptera: Scelionidae) [40,41,42,43]. Therefore, the major benefits, challenges, and recommendations for releasing those egg parasitoids in soybean and maize are discussed in this review.

## 2. *Trichogramma pretiosum*: Biology, Parasitism Capacity, and Major Release Recommendations

Several of the more than 200 recorded species of the genus *Trichogramma* (Hymenoptera: Trichogrammatidae) [44] have been successfully used in ABC programs against a wide range of lepidopteran pests worldwide [45]. Among those biocontrol agents, *Tr. pretiosum* is one of the most important in the Neotropics. *Trichogramma pretiosum* is a tiny egg parasitoid (~0.5 mm long) that has been extensively released in Central and South America and, to a lesser extent, in North America and Asia [45]. The development cycle (Figure 2) of this parasitoid can vary depending upon the host and temperature. Under 25 °C it takes ~10 days to develop from egg to adult when developing in the eggs of *Anticarsia gemmatalis* Hübner, 1818 (Lepidoptera: Eribidae), *Chysodeixis includens* (Walker, 1857) (Lepidoptera: Noctuidae) [41], and *S. frugiperda* [42]. Parasitism can be easily recognized in the field as darkened eggs (Figure 2) due to the accumulation of urate salts in the chorion of the parasitized eggs, which stays on the chorion of such eggs even after the parasitoid’s emergence [46].

In soybean cropped in the Neotropical region, *Tr. pretiosum* is responsible for more than 90% of the natural parasitism of lepidopteran eggs [47,48]. A single *Tr. pretiosum* female can parasitize more than 50 eggs of the most important Lepidoptera pests of soybean and maize during its lifespan (Table 1), illustrating the potential of the parasitoid as an applied biocontrol agent. Consequently, this egg parasitoid has been extensively reared and released in the fields of several countries in the Neotropical region. For instance, Brazil has 10 different companies rearing and selling this parasitoid to farmers. The recommendations on how to use this parasitoid vary depending on the target species (Table 2).

In addition to the recommendations on how to use this parasitoid described in Table 2, the following precautions are also strongly advised to ensure the successful use of *Tr. pretiosum.*

(1) *Trichogramma* spp. are usually released in the field as pupae close to adult emergence, due to the ease of transporting and handling pupae. These parasitoid pupae inside host eggs are released loosely in bulk and are uniformly distributed over the crop. This process can even be mechanized and performed using drones, which can reduce costs [38,39,40,41,42,43,44,45,46,47,48,49,50,51,52,53,54,55,56].

*Trichogramma* release must be carried out during the cooler hours of the day, especially in tropical countries like Brazil, in periods without rain or strong winds (especially when released with drones). Pupae should be very close to adult emergence to reduce unnecessary exposure to factors that could cause parasitoid mortality. In extensively cropped areas, including soybean and maize, abiotic variables, such as adverse temperature and heavy rainfall [57], or biotic factors, such as predation of the released pupae caused by a diverse variety of predators (ants, ladybugs, among others) [58], are among the most important causes of mortality of recently released pupae of this parasitoid [45,46,47,48,49,50,51,52,53,54,55,56,57,58,59]. Failure of ABC programs using *Trichogramma* spp. is frequently caused by high mortality of the biocontrol agent immediately after release [60,61].

**Table 2 insects-15-00869-t002:** Officially registered biological control products marketed in Brazil based on *Trichogramma pretiosum* to be released in the country followed by official recommendations from the producer companies [62].

Biological Target	Release Timing	Parasitoid Density/ha	Points per ha	Releases per Season	Interval of Releases	Commercial Brand (Producer Company and Registration Number in Brazil)
*Anticarsia gemmatalis*	When the presence of moths is visually observed in the area	500 thousand (soybean vegetative stage) 750 thousand (soybean reproductive stage)	50	2	4 days	Hunter (Koppert, 10,115); Tricho-VIT (JB Biotecnologia, 29,118); Pretiobug (CP2 Ltda, 2315); TrichoAgri (IBI Agentes Biológicos, 16,517); Trichogramma (AMIPA, 40,517); Trichobio-P (Farmbio, 6619); Trichomip-P (Promip, 8815); Trichosul (Sul-Mip, 20,220); Trilag (Topbio, 29,418); BioIn-Tricho-P (BioIn 32,621).
*Chrysodeixis includens*	When the presence of moths is visually observed in the area	500 thousand (soybean vegetative stage) 750 thousand (soybean reproductive stage)	50	2	4 days	
*Spodoptera frugiperda*	When 3 *S. frugiperda* moths are captured per pheromone trap (install 1 trap per 5 hectares)	100 thousand	25	3	7 days	
*Helicoverpa zea*	20% emission of styles and stigmas (plants)	100 thousand	25	3	3 to 7 days	

Alternatively, parasitoids can be released inside biodegradable capsules along with honey to feed the adults immediately after emergence [63]. This allows for the release of fed adults rather than pupae near emergence. The release of parasitoids in capsules can also be carried out by drones, but this reduces the operational capacity of applications due to the large volume of capsules to be released. On the one hand, the increased operational time required for the release and the need to provide food for the adults make releasing adults more expensive, which can discourage farmers from adopting the technology. On the other hand, releasing fed adults inside capsules can reduce the risks of *Trichogramma* spp. pupae mortality and improve overall parasitism performance.

Parasitoids with access to a food source in the field have a longer lifespan and a higher parasitism rate than parasitoids exposed to food deprivation [64]. Furthermore, once mature eggs are depleted, parasitoid females with access to honey are also reported to contribute to greater non-reproductive host mortality [65]. In addition, the release of fed *Tr. pretiosum* adults could allow farmers to delay the parasitoid release due to unfavorable weather conditions (rainy or extremely hot days), if necessary, for a couple of days after the adults emerge. Waiting for more favorable weather conditions to release the parasitoids can reduce their mortality due to unfavorable weather conditions. For instance, Roswadoski [63] recorded a 15-day increase in the lifespan of another egg parasitoid species (*Telenomus podisi*) inside capsules with honey without harming the efficiency of the parasitoid in controlling pests.

Releasing fed adults inside capsules is more expensive; however, considering the extensive areas cultivated with soybean and maize and the large continuous fields with those crops in some countries, this release technology may be a necessary solution to enable the large-scale use of these macroorganisms. Regardless of the release technology (pupae or fed adults), the release should not be carried out during periods with high temperatures, heavy rain, or strong winds.

(2) *Trichogramma pretiosum* should only be released in fields where IPM is adopted. IPM provides a more balanced environment due to the reduced use of chemical insecticides, always prioritizing products with a higher selectivity toward the parasitoid and other natural antagonists [13]. This more balanced environment will provide greater survival and success of the released parasitoids throughout the crop season [8]. Traditional synthetic chemical products (especially insecticides) should not be applied on the crop for at least 10 days before and five days after the release of *Trichogramma* spp. [47]. When insecticides are inevitable, it is important to use the most selective options available, which include other bioinsecticides or insect growth regulators [13].

## 3. *Trichogramma pretiosum* Field Results

In the Brazilian soybean season of 2013/14, an ABC pilot program compared the release of *Tr. pretiosum* pupae in IPM soybean fields with conventional chemical insecticide fields throughout the state of Paraná in southern Brazil [66]. A total of 19 soybean fields (different farms) were managed by releasing *Tr. pretiosum* pupae inside fields where IPM was adopted. Parasitoid releases began around three days after the visual detection of the first moths of the target species (Table 2) or their capture by light traps installed in the areas (considered early Lepidoptera infestation, the ideal time to release *Tr. pretiosum*). Two releases were carried out per area in weekly intervals, totaling 100,000 parasitoids per hectare in each release. Whenever the economic threshold of 30% defoliation in the vegetative soybean stage or 15% in the reproductive stage was reached, the soybean field was sprayed with a chemical control chosen by the farmer [66]. In the results obtained, fewer sprays of chemical insecticides in the fields occurred where IPM was adopted with releases of *Tr. pretiosum* than in the fields where traditional insecticides were sprayed according to the farmer’s decision (Table 3). Thus, the benefits of releasing *Tr. pretiosum* within IPM context were economically and environmentally positive [66].

Similar results were also recorded by Basso et al. [67] after releasing 200,000 *Tr. pretiosum* divided into two terrestrial releases (100,000 parasitoids in each release) 20 days apart or four weekly releases (50,000 parasitoids in each release) performed by drone. In both reports [66,67], the use of *Tr. pretiosum* in soybean was an efficient and viable management alternative, helping reduce the use of chemical insecticides. The release of *Tr. pretiosum* associated with the adoption of IPM ensured a good yield with a reduced environmental impact, an extremely important demand for sustainable soybean production [3]. In this context, the use of *Tr. pretiosum* proved to be efficient and a good alternative for soybean farmers adopting IPM.

In maize, *S. frugiperda* is one of the most important pest species in many countries [68]. *Spodoptera frugiperda* is not easily controlled using only *Trichogramma* spp. because most females of *Trichogramma* spp. can only access the upper layer of egg masses and cannot easily oviposit through the scales left by *S. frugiperda* moths covering their egg masses (which act as a physical defense against parasitism) [40,69,70]. Despite such challenges, after three releases of 100,000 *Tr. pretiosum* each in weekly intervals, a substantial economic gain of USD 96.48 ha^−1^ (19.4%) was recorded in maize yield [71]. The timing of the first parasitoid release was based on reaching an action limit of three or more moths (cumulatively) captured per pheromone trap installed in the area [72].

Furthermore, *Tr. pretiosum* can contribute to the management of different Lepidoptera species, not only on non-*Bt* crops but also on *Bt* soybean and maize, to mitigate outbreaks of those pests (Table 1). Therefore, the use of *Tr pretiosum* is a more sustainable option than using chemical insecticides to control *Bt* soybean resistant species (*Rachiplusia nu*, *Crosidosema* sp.) or *Bt* maize-resistant *S. frugiperda* populations, which leads to undesired increases in insecticide use [52].

## 4. *Telenomus remus:* Biology, Parasitism Capacity, and Major Release Recommendations

Similarly to *Tr. pretiosum*, *Te. remus* is also a small-sized egg parasitoid (Figure 3), measuring from 0.5 to 0.6 mm in length, and has a short life cycle. This parasitoid was introduced into the Americas as a biological control agent for managing *Spodoptera* spp. [73]. *Telenemus remus* has been widely studied and released in various countries against *S. frugiperda* and other species of the genus *Spodoptera* [40,41,42,43,44,45,46,47,48,49,50,51,52,53,54,55,56,57,58,59,60,61,62,63,64,65,66,67,68,69,70,71,72,73,74], such as *S. eridania* and *S. cosmioides*, which are important pests in crops including cotton and soybean [75,76].

The potential of *Te. remus* as a biological control agent stands out mainly due to its high parasitism capacity (Table 4), especially on eggs of *Spodoptera* spp. [40]. Many of these pests lay their eggs in overlapping layers, covered with scales from the moth’s wings, which usually provide a protective barrier against parasitism; however, adults of *Te. remus* can overcome this [40,41,42,43,44,45,46,47,48,49,50,51,52,53,54,55,56,57,58,59,60,61,62,63,64,65,66,67,68,69,70,71,72,73,74,75,76,77]. In addition, *Te. remus* has a high capacity for dispersal in the field [78] and a strong searching ability [79]. An adult female can parasitize an average of 121.05 *S. frugiperda* eggs in just the first 24 h of parasitism [42], and up to 220 eggs throughout the parasitoid’s lifetime [80], which lasts on average between 8 and 13 days depending upon the temperature (Table 4). These characteristics illustrate the high potential of this parasitoid for ABC programs against *Spodoptera* spp.

Despite its high control potential, the use of *Te. remus* to manage *S. frugiperda* is still limited due to higher production costs than *Tr. pretiosum*. These higher costs are because its rearing is more labor intensive and, therefore, more expensive. Furthermore, *Te. remus* is a more specific parasitoid for *Spodoptera* spp. eggs, having a smaller host range than *Tr. pretiosum* [40].

To successfully release *Te. remus* in soybean or maize, the following recommendations are important.

(1) The release of fed adults is necessary for *Te. remus* because, unlike *Tr. pretiosum*, male *Te. remus* emerge up to 24 h before the females. Therefore, no matter how close to adult emergence the release of pupae is carried out, the females will be exposed to the weather conditions and predators in the field for at least 24 h, which can significantly increase the mortality of the released pupae, compromising the management strategy [40]. Despite being necessary, releasing fed *Te. remus* adults inside capsules, as previously discussed for *Tr. pretiosum*, may increase release costs [63], discouraging the use of this parasitoid species.

(2) Similarly to *Tr. pretiosum*, the release of *Te. remus* should be carried out in soybean or maize fields with the adoption of IPM. This is essential to prevent the released parasitoids from being significantly eliminated from the field by unnecessary chemical insecticide applications using non-selective insecticides. The use of traditional chemical insecticides or other chemicals should be avoided for at least 10 days before and 7 days after the release of the parasitoid. When insecticides are inevitable, the most selective ones should be prioritized [13]. The greater stability of the environment observed where IPM is adopted is essential for the success of the biocontrol agent [8].

## 5. *Telenomus remus* Field Results

The results of ABC programs adopting *Te. remus* have vary significantly [40]. In Brazil, no positive results were recorded in field conditions [81]. However, after the release of *Te. remus* in maize fields in Florida (USA), 43% parasitism of eggs of *S. frugiperda* was recorded [82]. Better results varying from parasitism of 60% to 100% were reported from Colombia [83], Guyana, Suriname [84,85], Venezuela [86], and Barbados [87]. In Honduras, the reported parasitism of *S. frugiperda* eggs after the release of *Te. remus* varied from 20% to 92% [87].

Variations in the results recorded in those studies might be due to different parasitoid strains, the number of parasitoids released, the stage of the parasitoid adopted in the release (adults or pupae), and/or the number of releases (Table 4), or due to the weather conditions in the fields during releases. A positive effect of increased humidity on *Te. remus* parasitism has been reported from laboratory studies in *S. litura* [88], *Agrotis spinifera* (Hubner, 1808) (Lepidoptera: Noctuidae) [89], and *Corcyra cephalonica* (Stainton, 1865) (Lepidoptera: Pyralidae) eggs [90]. Similar effects of humidity were also reported for *Telenomus isis* (Polaszek, 1993) (Hymenoptera: Scelionidae) parasitizing coffee borer eggs [91]. Despite the effect of humidity, the effects of environmental conditions can differ among parasitized host species [92].

## 6. Associating *Telenomus remus* with *Trichogramma pretiosum* for the Management of Lepidopteran Pests in Soybean and Maize Crops

Both egg parasitoid species discussed previously have specificities and, therefore, advantages and disadvantages. While *Te. remus* has a higher parasitism capacity in *Spodoptera* spp. eggs than *Tr. pretiosum*, the latter has a much wider host range and a lower rearing cost. However, although *Tr. pretiosum* parasitize *S. frugiperda* eggs, isolated use of this parasitoid to manage *S. frugiperda* in maize or soybean faces challenges. It is frequently not successful enough to eliminate the need for additional management strategies. When evaluating the exclusive use of the parasitoid species to manage *S. frugiperda* in maize fields, *Tr. pretiosum* reached 25% parasitism, while *Te. remus* reached 57% of parasitism [93], confirming what is described in the literature, that the exclusive use of *Tr. pretiosum* will not provide good results in managing *S. frugiperda,* and highlighting the potential of *Te. remus* as an effective biological control agent of this pest. Thus, a possible alternative could be an association between both parasitoids to combine their benefits, reducing the weakness of the isolated use of a single parasitoid species [93]. The association of different parasitoid species has been scarcely tested in the laboratory and even less in the field to understand the effects and potential of combining egg parasitoids for managing *S. frugiperda* [94,95,96] and other pest species [97] with some positive results. Additional effects are expected from controlling an increased number of target species besides preserving a higher biodiversity of natural biocontrol agents in the agroecosystem. The combination of both *Tr. pretiosum* and *Te remus* could reduce the costs of the exclusive use of *Te. remus* and still increase the broad-spectrum control of parasitoid releases [96].

Despite such theoretical benefits, when further studied in maize fields, the combined release of *Tr. pretiosum* and *Te. remus* did not increase the parasitism of *S. frugiperda* eggs compared to the isolated release of *Te. remus*; however, their releases did not decrease *Te. remus* parasitism [93,94,95,96]. In this context, further studies about the association of *Tr. pretiosum* and *Te. remus* are still necessary. Considering that *S. frugiperda* can occur together with other lepidopterans and increasing the biodiversity of the natural biological control population is usually positive, the combination of *Tr. pretiosum* and *Te. remus* could still be an alternative and more sustainable approach to pest management.

Farmers typically face challenges with the occurrence of multiple pest species at the same time in their crops [96]. It is important to consider that the possible mixture of parasitoids does not necessarily need to result in more parasitism than releasing just one species. If the mixture of parasitoids is sufficiently effective, it could be an option to maintain the same rate of parasitism but increase the biodiversity of parasitoids, which can be an important goal for a more balanced agroecosystem [96].

## 7. Final Considerations

The use of *Tr. pretiosum* is already a reality in soybean and maize fields after the commercial registration of this egg parasitoid species in Brazil. However, no commercial product with *Te. remus* has been registered and marketed in the country, despite it being a more efficient egg parasitoid for controlling *Spodoptera* complex than *Tr. pretiosum*. Therefore, the official registration and marketing of *Te. remus* for the management of *Spodoptera* spp. would be interesting and benefit farmers when complete.

Egg parasitoids (*Tr. pretiosum* and *Te. remus*) are very efficient management options and can be released to reduce or even eliminate the use of chemical insecticides against certain target pests (previously listed in Table 1) in both non-*Bt* and *Bt* fields. Therefore, these parasitoids can be a very important alternative strategy to manage lepidopteran populations resistant to *Bt* proteins, stopping the increasing use of insecticides for this. However, for the successful adoption of egg parasitoids (both *Tr. pretiosum* and *Te. remus*), it is necessary to properly adopt the technology as detailed in this review in fields following IPM recommendations. Integrating the use of the egg parasitoids with other compatible pest control strategies inside the IPM framework is crucial. Resistant plants, for instance *Bt* plants, as well as botanical insecticides, entomopathogens and even selective chemical insecticides among other environmentally friendly control strategies are essential to complement pest control while preserving the released egg parasitoids in the agroecosystem. This will help to maintain the egg parasitoids in the area and, therefore, reduce the dependency of new releases in each pest cycle.

## Figures and Tables

**Figure 1 insects-15-00869-f001:**
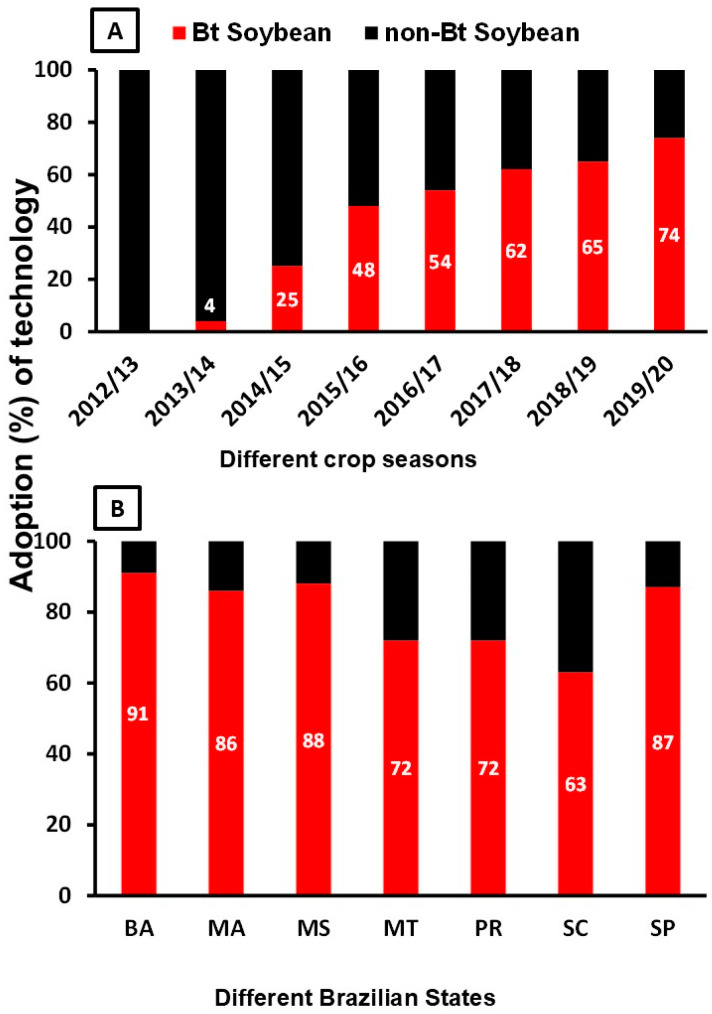
Adoption of *Bt* soybeans in Brazil (%) over the years (**A**) and in different states of Brazil in the 2019/20 season (**B**). Adapted from Bueno & Silva [31]. Brazilian States: Bahia (BA), Maranhão (MA), Mato Grosso do Sul (MS), Mato Grosso (MT), Paraná (PR), Santa Catarina (SC), and São Paulo (SP).

**Figure 2 insects-15-00869-f002:**
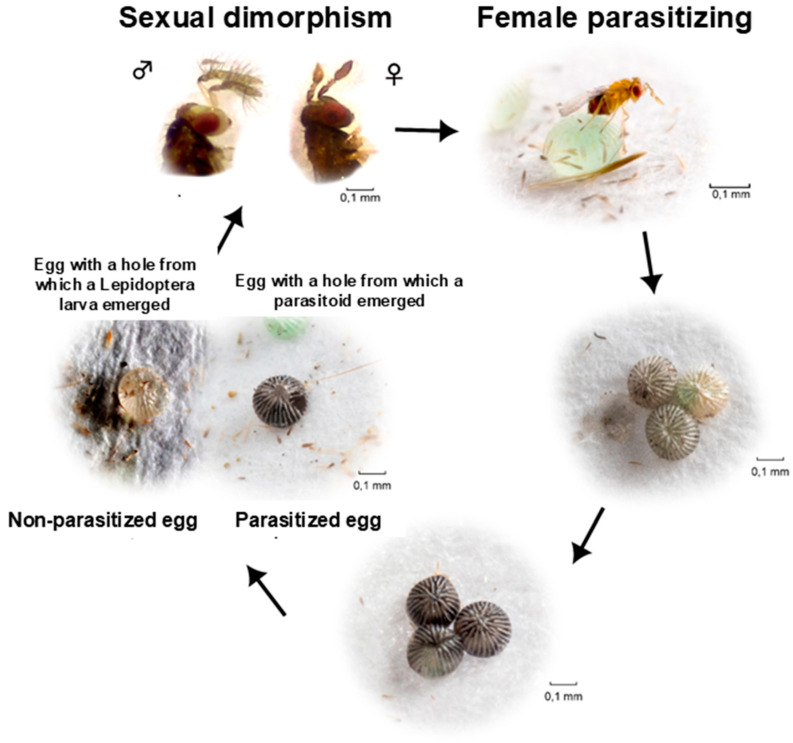
Life cycle of *Trichogramma pretiosum* parasitizing *Anticarsia gemmatalis* eggs. Adapted from Bueno et al. [47].

**Figure 3 insects-15-00869-f003:**
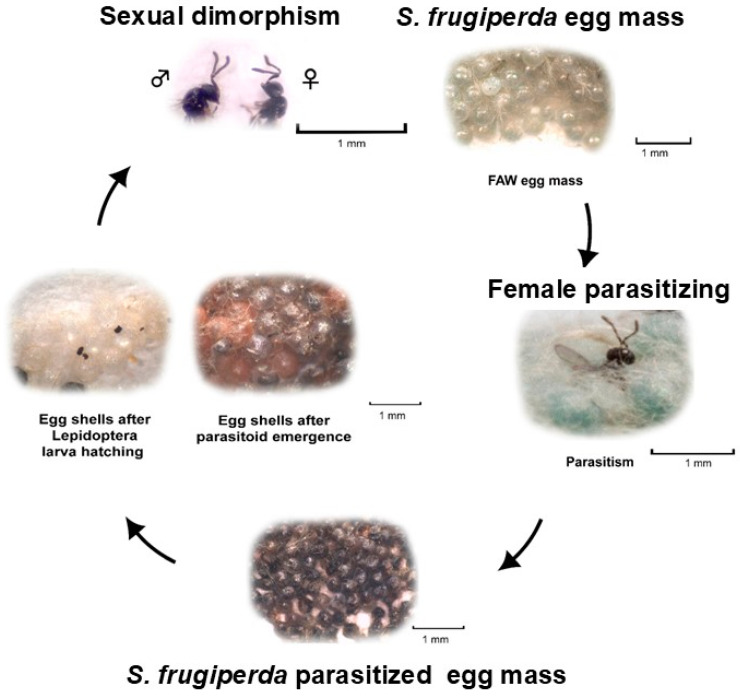
Life cycle of *Telenomus remus* parasitizing *Spodoptera frugiperda* eggs. Adapted from Colmenarez et al. [40].

**Table 1 insects-15-00869-t001:** Parasitism capacity of *Trichogramma pretiosum* at 25 °C on different host eggs.

Host Specie	Lifetime Parasitism (Number of Parasitized Eggs/Female)	Longevity of Parental Females (Days)	Reference
*Spodoptera frugiperda*	14.8	9.8	[42]
*Spodoptera cosmioides*	51.7	5.6	[49]
*Spodoptera eridania*	19.9	- ^1^	[50]
*Anticarsia gemmatalis*	33.0	5.0	[51]
*Crysodeixis includens*	40.9	10.1	[51]
*Rachiplusia nu*	46.4	10.5	[52]
*Trichoplusia ni*	53.0	10.1	[53]
*Helicoverpa armigera*	17.4 to 85.0	12.3	[54,55]
*Helicoverpa zea*	18.7	6.0	[55]
*Chloridea virescens*	23.7	7.4	[55]

^1^ - Not evaluated.

**Table 3 insects-15-00869-t003:** Results from farms adopting soybean-IPM with the release of *Trichogramma pretiosum* and farms not adopting soybean-IPM. Londrina, Paraná, Brazil. Crop Season 2013/14 [66].

Pest Management ^1^	Number of Sprays	Days Until First Insecticide Spray	Costs of Control US$/ha			Cost ^4^ (kg/ha)	Yield (kg/ha)
			Inputs ^2^	Service ^3^	Total		
IPM+Tricho	2.05	61	15.5	10.1	25.6	144	2903.4
Non-IPM	4.99	-	31.9	22.1	54.0	300	2920.2

**^1^** IPM+Tricho: Integrated pest management fields with 1 release of *Trichogramma pretiosum* (19 fields); Non-IPM: Soybean fields which did not follow IPM recommendations (conventional farmer management) (333 farmers). **^2^** Insecticides USD 9.10/ha and *T. pretiosum* USD 6.07/ha. **^3^** Insecticide spray USD 9.08/ha and workers to manually release *T. pretiosum* USD 1.04/ha. **^4^** Total costs transformed into equivalent value of soybean price in that year. USD/BRL exchange rate of 5.6 dollars/1.0 real.

**Table 4 insects-15-00869-t004:** Parasitism capacity of *Telenomus remus* at 25 °C in different host eggs. Adapted from Colmenarez et al. [40].

Host Species	Lifetime Parasitism (Number of Parasitized Eggs/Female)	Longevity of Parental Females (Days)	Reference
*Spodoptera frugiperda*	140.8 to 220.0	8.3 to 10.6	[79,80]
*Spodoptera cosmioides*	115.3	13.1	[79]
*Spodoptera eridania*	139.5	8.0	[79]
*Anticarsia gemmatalis*	200.5	12.4	[80]

## Data Availability

No new data were created or analyzed in this review article. Data sharing is not applicable.

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
