# Peer review of "Using Egg Parasitoids to Manage Caterpillars in Soybean and Maize: Benefits, Challenges, and Major Recommendations"

_insects, 2024, doi:10.3390/insects15110869_

Round 1

Reviewer 1 Report

Comments and Suggestions for Authors

i have the feeling that you need to connect better the ideas regarding biology and suggestions in the use.  I think you make big recommendation in gral and not specific for the crops in X area, just in general, I would like to read more specific examples.

Author Response

Dear Reviewer, 

Considering that the time of both editor and referees is extremely precious and given on a voluntary basis and, therefore, aiming at saving time at the second round of evaluation, we highlighted the most substantial changes/considerations made due to reviewer 1 in blue; reviewer 2 in green; and reviewer 3 in red in the new version of the manuscript and detailed explanation of the modifications done is the attached file.

Sincerely,

Adeney de Freitas Bueno, author for correspondence.

Reviewer 2 Report

Comments and Suggestions for Authors

The review paper by Bueno et al., provides a comprehensive discussion about the use of parasitoids in Brazil. The focus of this review comprises two parasitoid species. The review is well-written, discussing benefits, challenges, and general recommendations.  I have made several suggestions and comments to the attached file.

Comments on the Quality of English Language

The English will need some improvement, as some sentences are hard to understand.

Author Response

Dear Reviewer, 

Considering that the time of both editor and referees is extremely precious and given on a voluntary basis and, therefore, aiming at saving time at the second round of evaluation, we highlighted the most substantial changes/considerations made due to reviewer 1 in blue; reviewer 2 in green; and reviewer 3 in red in the new version of the manuscript and detailed explanation of the modifications done is the attached file.

Sincerely,

Adeney de Freitas Bueno, author for correspondence

Reviewer 3 Report

Comments and Suggestions for Authors

A brief summary

Egg parasitoids are one component of the Integrated Pest Management that many found as an effective strategy in managing lepidopteran pests. The authors of this manuscript conducted extensive literature reviews on the use of egg parasitoids in managing lepidopteran larval pests in soybean and maize with focus on two parasitoids: Trichogramma pretiosum and Telenomus remus. From the abstract, introduction, sections on Tr. pretiosum and Te. remus, to the final considerations, the manuscript was well-written and easy to follow. Some improvements (including grammar) can be made to strengthen the content of this review manuscript – see specific comments below.

Specific comments:

Below are a few comments and/or suggestions to be addressed. I only refer to the line numbers since the page numbering was not correct in some pages (I.e. same page numbers were repeated twice).

Simple summary

Line 19

In this study … à Since this is not a research paper,  this phrase should be corrected to: “In this review article …” or along this line.

Introduction

Lines 53-79

There are 2 different font size in this section. Be sure font size is consistent throughout the document and follow the journal guidelines.

Figure 1

Line 95

Although you mentioned in the body of the introduction, it is important to provide information (spell out) on the abbreviated state. They can be listed after “…2019/2020 season (B) [31]”

Section 2

Figure 2

Line 135

Under sexual dimorphism: egg with a hole from which a Lepidoptera larvae … à incorrect grammar. It should be a lepidopteran larva OR a Lepidoptera larva OR lepidopteran larvae OR Lepidoptera larvae.

Line 154

… drones what can reduce … à should be: … drones which can reduce …

Line 161

… such as predation of the released pupae … à Predation by what? It should be expanded a little bit to explaining the predation.

Table 2

Lines 167-169

I think it would be helpful for readers if you provide a clear separation to show which company that produce which biological targets. As of now, this was not very clear. For example, Trichomp-P (Promip, 8815): is it for Spodoptera frugiperda or Helicoverpa zea?

Under ‘parasitoid density’ column: the first two lines showed 2 different densities for the same soybean stage. Is this correct??

Missing a bracket: under ‘release timing’ for S. frugiperda

Section 3

Lines 217-218

The sentence needs to be rewritten. As of now it tells me that farmers sprayed chemical control. Was it chemical control or soybean field that was sprayed?

Table 3

Line 223

… and farms which did not adopted … à grammatical error, it should be: … and farms not adopting

Line 228

… for each real.” à This seems unfinished sentence: the word ‘real’ is out of place.

Section 6

Line  326

When evaluated  the … à grammatical error. It should be” When evaluating the …

Section 7

You mentioned the needs to do in the scheme IPM. It will be good to add into discussions other components of IPM that can strengthen the adoption of parasitoids. Consider something like the use of soybean and maize that are resistant to lepidopteran pests, or other techniques such as microbial control.

In the earlier sections you indicated that it is not easy for farmer to adopt egg parasitoids technology. It will be also good to add references on efforts that has been done to educate these farmers in regard to the technology.

Comments on the Quality of English Language

See above comments on some grammatical errors. Although overall quality of English is acceptable, it would be good to consider having an English native speaker reviewing your manuscript.

Author Response

(The authors gave the same response as above.)

Round 2

Reviewer 2 Report

Comments and Suggestions for Authors

The authors have addressed all the questions and concerns appropriately. The manuscript can be accepted for publication in its current form.